# No-Regret Learning in Unknown Games with Correlated Payoffs

**Pier Giuseppe Sessa**
ETH Zürich
sessap@ethz.ch

**Ilija Bogunovic**
ETH Zürich
ilijab@ethz.ch

**Maryam Kamgarpour**
ETH Zürich
maryamk@ethz.ch

**Andreas Krause**
ETH Zürich
krausea@ethz.ch

## Abstract

We consider the problem of learning to play a repeated multi-agent game with an *unknown* reward function. Single player online learning algorithms attain strong regret bounds when provided with full information feedback, which unfortunately is unavailable in many real-world scenarios. Bandit feedback alone, i.e., observing outcomes only for the selected action, yields substantially worse performance. In this paper, we consider a natural model where, besides a noisy measurement of the obtained reward, the player can also observe the opponents' actions. This feedback model, together with a regularity assumption on the reward function, allows us to exploit the correlations among different game outcomes by means of Gaussian processes (GPs). We propose a novel confidence-bound based bandit algorithm GP-MW, which utilizes the GP model for the reward function and runs a multiplicative weight (MW) method. We obtain novel *kernel-dependent regret bounds* that are comparable to the known bounds in the full information setting, while substantially improving upon the existing bandit results. We experimentally demonstrate the effectiveness of GP-MW in random matrix games, as well as real-world problems of traffic routing and movie recommendation. In our experiments, GP-MW consistently outperforms several baselines, while its performance is often comparable to methods that have access to full information feedback.

## 1 Introduction

Many real-world problems, such as traffic routing [14], market prediction [10], and social network dynamics [21], involve multiple learning agents that interact and compete with each other. Such problems can be described as *repeated games*, in which the goal of every agent is to maximize her cumulative reward. In most cases, the underlying game is *unknown* to the agents, and the only way to learn about it is by repeatedly playing and observing the corresponding game outcomes.

The performance of an agent in a repeated game is often measured in terms of *regret*. For example, in traffic routing, the regret of an agent quantifies the reduction in travel time had the agent known the routes chosen by the other agents. No-regret algorithms for playing unknown repeated games exist, and their performance depends on the information available at every round. In the case of *full information* feedback, the agent observes the obtained reward, as well as the rewards of other non-played actions. While these algorithms attain strong regret guarantees, such full information feedback is often unrealistic in real-world applications. In traffic routing, for instance, agents only observe the incurred travel times and cannot observe the travel times for the routes not chosen.

In this paper, we address this challenge by considering a more realistic feedback model, where at every round of the game, the agent plays an action and observes the noisy reward outcome. In addition to this *bandit* feedback, the agent *also observes the actions played by other agents*. Under this feedback model and further regularity assumptions on the reward function, we present a novel

|  | HEDGE [11] | EXP3 [3] | GP-MW [this paper] |
|---|---|---|---|
| **Feedback** | rewards for all actions | obtained reward | obtained reward + opponents' actions |
| **Regret** | $\mathcal{O}\big(\sqrt{T \log K_i}\big)$ | $\mathcal{O}\big(\sqrt{T K_i \log K_i}\big)$ | $\mathcal{O}\big(\sqrt{T \log K_i} + \gamma_T \sqrt{T}\big)$ |

Table 1: Finite action set regret bounds that depend on the available feedback observed by player $i$ at each time step. Time horizon is denoted with $T$, and $K_i$ is the number of actions available to player $i$. Kernel dependent quantity $\gamma_T$ (Eq. (3)) captures the degrees of freedom in the reward function.

no-regret algorithm for playing unknown repeated games. The proposed algorithm alleviates the need for full information feedback while still achieving comparable regret guarantees.

**Related Work.** In the *full information setting*, multiplicative-weights (MW) algorithms [17] such as HEDGE [11] attain optimal $\mathcal{O}\big(\sqrt{T \log K_i}\big)$ regret, where $K_i$ is the number of actions available to agent $i$. In the case of convex action sets in $\mathbb{R}^{d_i}$, and convex and Lipschitz rewards, online convex optimization algorithms attain optimal $\mathcal{O}\big(\sqrt{T}\big)$ regret [25]. By only assuming Lipschitz rewards and bounded action sets, $\mathcal{O}\big(\sqrt{d_i T \log T}\big)$ regret follows from [18], while in [13] the authors provide efficient gradient-based algorithms with 'local' regret guarantees. Full information feedback requires perfect knowledge of the game and is unrealistic in many applications. Our proposed algorithm overcomes this limitation while achieving comparable regret bounds.

In the more challenging *bandit setting*, existing algorithms have a *substantially worse* dependence on the size of the action set. For finite actions, EXP3 [3] and its variants ensure optimal $\mathcal{O}\big(\sqrt{T K_i \log K_i}\big)$ regret. In the case of convex action sets, and convex and Lipschitz rewards, bandit algorithms attain $\mathcal{O}(\mathrm{poly}(d_i)\sqrt{T})$ regret [6], while in the case of Lipschitz rewards $\mathcal{O}\big(T^{\frac{d_i+1}{d_i+2}} \log T\big)$ regret can be obtained [22]. In contrast, our algorithm works in the *noisy* bandit setting and requires the knowledge of the actions played by other agents. This allows us to, under some regularity assumptions, obtain substantially improved performance. In Table 1, we summarize the regret and feedback model of our algorithm together with the existing no-regret algorithms.

The previously mentioned online algorithms reduce the unknown repeated game to a single agent problem against an adversarial and adaptive environment that selects a different reward function at every time step [7]. A fact not exploited by these algorithms is that in a repeated game, the rewards obtained at different time steps are *correlated* through a static unknown reward function. In [24] the authors use this fact to show that, if every agent uses a regularized no-regret algorithm, their individual regret grows at a lower rate of $\mathcal{O}(T^{1/4})$, while the sum of their rewards grows only as $\mathcal{O}(1)$. In contrast to [24], we focus on the single-player viewpoint, and we do not make any assumption on opponents strategies[1]. Instead, we show that by observing opponents' actions, the agent can exploit the structure of the reward function to reduce her individual regret.

**Contributions.** We propose a novel no-regret bandit algorithm GP-MW for playing unknown repeated games. GP-MW combines the ideas of the multiplicative weights update method [17], with GP upper confidence bounds, a powerful tool used in GP bandit algorithms (e.g., [23, 5]). When a finite number $K_i$ of actions is available to player $i$, we provide a novel high-probability regret bound $\mathcal{O}(\sqrt{T \log K_i} + \gamma_T \sqrt{T})$, that depends on a kernel-dependent quantity $\gamma_T$ [23]. For common kernel choices, this results in a sublinear regret bound, which grows only logarithmically in $K_i$. In the case of infinite action subsets of $\mathbb{R}^{d_i}$ and Lipschitz rewards, via a discretization argument, we obtain a high-probability regret bound of $\mathcal{O}(\sqrt{d_i T \log(d_i T)} + \gamma_T \sqrt{T})$. We experimentally demonstrate that GP-MW outperforms existing bandit baselines in random matrix games and traffic routing problems. Moreover, we present an application of GP-MW to a novel robust Bayesian optimization setting in which our algorithm performs favourably in comparison to other baselines.

## 2  Problem Formulation

We consider a repeated static game among $N$ non-cooperative agents, or players. Each player $i$ has an action set $\mathcal{A}^i \subseteq \mathbb{R}^{d_i}$ and a reward function $r^i : \mathcal{A} = \mathcal{A}^1 \times \cdots \times \mathcal{A}^N \to [0, 1]$. We assume that the reward function $r^i$ is unknown to player $i$. At every time $t$, players simultaneously choose actions $\mathbf{a}_t = (a_t^1, \ldots, a_t^N)$ and player $i$ obtains a reward $r^i(a_t^i, a_t^{-i})$, which depends on the played action $a_t^i$

and the actions $a_t^{-i} := (a_t^1, \ldots, a_t^{i-1}, a_t^{i+1}, \ldots, a_t^N)$ of all the other players. The goal of player $i$ is to maximize the cumulative reward $\sum_{t=1}^T r^i(a_t^i, a_t^{-i})$. After $T$ time steps, the *regret* of player $i$ is defined as

$$R^i(T) = \max_{a \in \mathcal{A}^i} \sum_{t=1}^T r^i(a, a_t^{-i}) - \sum_{t=1}^T r^i(a_t^i, a_t^{-i}), \qquad (1)$$

i.e., the maximum gain the player could have achieved by playing the single best fixed action in case the sequence of opponents' actions $\{a_t^{-i}\}_{t=1}^T$ and the reward function were known in hindsight. An algorithm is *no-regret* for player $i$ if $R^i(T)/T \to 0$ as $T \to \infty$ for any sequence $\{a_t^{-i}\}_{t=1}^T$.

First, we consider the case of a finite number of available actions $K_i$, i.e., $|\mathcal{A}^i| = K_i$. To achieve no-regret, the player should play mixed strategies [7], i.e., probability distributions $\mathbf{w}_t^i \in [0,1]^{K_i}$ over $\mathcal{A}^i$. With full-information feedback, at every time $t$ player $i$ observes the vector of rewards $\mathbf{r}_t = [r^i(a, a_t^{-i})]_{a \in \mathcal{A}^i} \in \mathbb{R}^{K_i}$. With bandit feedback, only the reward $r^i(a_t^i, a_t^{-i})$ is observed by the player. Existing full information and bandit algorithms [11, 3], reduce the repeated game to a sequential decision making problem between player $i$ and an adaptive environment that, at each time $t$, selects a reward function $r_t : \mathcal{A}_i \to [0,1]$. In a repeated game, the reward that player $i$ observes at time $t$ is a *static* fixed function of $(a_t^i, a_t^{-i})$, i.e., $r_t(a_t^i) = r^i(a_t^i, a_t^{-i})$, and in many practical settings similar game outcomes lead to similar rewards (see, e.g., the traffic routing application in Section 4.2). In contrast to existing approaches, we exploit such *correlations* by considering the feedback and reward function models described below.

**Feedback model.** We consider a *noisy* bandit feedback model where, at every time $t$, player $i$ observes a noisy measurement of the reward $\tilde{r}_t^i = r^i(a_t^i, a_t^{-i}) + \epsilon_t^i$ where $\epsilon_t^i$ is $\sigma_i$-sub-Gaussian, i.e., $\mathbb{E}[\exp(c\,\epsilon_t^i)] \leq \exp(c^2\sigma_i^2/2)$ for all $c \in \mathbb{R}$, with independence over time. The presence of noise is typical in real-world applications, since perfect measurements are unrealistic, e.g., measured travel times in traffic routing.

Besides the standard noisy bandit feedback, we assume player $i$ also observes the played actions $a_t^{-i}$ of all the other players. In some applications, the reward function $r^i$ depends only indirectly on $a_t^{-i}$ through some aggregative function $\psi(a_t^{-i})$. For example, in traffic routing [14], $\psi(a_t^{-i})$ represents the total occupancy of the network's edges, while in network games [15], it represents the strategies of player $i$'s neighbours. In such cases, it is sufficient for the player to observe $\psi(a_t^{-i})$ instead of $a_t^{-i}$.

**Regularity assumption on rewards.** In this work, we assume the unknown reward function $r^i : \mathcal{A} \to [0,1]$ has a bounded norm in a reproducing kernel Hilbert space (RKHS) associated with a positive semi-definite kernel function $k^i(\cdot, \cdot)$, that satisfies $k^i(\mathbf{a}, \mathbf{a}') \leq 1$ for all $\mathbf{a}, \mathbf{a}' \in \mathcal{A}$. The RKHS norm $\|r^i\|_{k^i} = \sqrt{\langle r^i, r^i \rangle_{k^i}}$ measures the smoothness of $r^i$ with respect to the kernel function $k^i(\cdot, \cdot)$, while the kernel encodes the similarity between two different outcomes of the game $\mathbf{a}, \mathbf{a}' \in \mathcal{A}$. Typical kernel choices are *polynomial*, *Squared Exponential*, and *Matérn*:

$$k_{poly}(\mathbf{a}, \mathbf{a}') = \left(b + \frac{\mathbf{a}^\top \mathbf{a}'}{l}\right)^n, \qquad k_{SE}(\mathbf{a}, \mathbf{a}') = \exp\left(-\frac{s^2}{2l^2}\right),$$

$$k_{Matérn}(\mathbf{a}, \mathbf{a}') = \frac{2^{1-\nu}}{\Gamma(\nu)}\left(\frac{s\sqrt{2\nu}}{l}\right)^\nu B_\nu\left(\frac{s\sqrt{2\nu}}{l}\right),$$

where $s = \|\mathbf{a} - \mathbf{a}'\|_2$, $B_\nu$ is the modified Bessel function, and $l, n, \nu > 0$ are kernel hyperparameters [20, Section 4]. This is a standard smoothness assumption used in kernelized bandits and Bayesian optimization (e.g., [23, 9]). In our context it allows player $i$ to use the observed history of play to learn about $r^i$ and predict unseen game outcomes. Our results are not restricted to any specific kernel function, and depending on the application at hand, various kernels can be used to model different types of reward functions. Moreover, composite kernels (see e.g., [16]) can be used to encode the differences in the structural dependence of $r^i$ on $a^i$ and $a^{-i}$.

It is well known that Gaussian Process models can be used to learn functions with bounded RKHS norm [23, 9]. A GP is a probability distribution over functions $f(\mathbf{a}) \sim \mathcal{GP}(\mu(\mathbf{a}), k(\mathbf{a}, \mathbf{a}'))$, specified by its mean and covariance functions $\mu(\cdot)$ and $k(\cdot, \cdot)$, respectively. Given a history of measurements $\{y_j\}_{j=1}^t$ at points $\{\mathbf{a}_j\}_{j=1}^t$ with $y_j = f(\mathbf{a}_j) + \epsilon_j$ and $\epsilon_j \sim \mathcal{N}(0, \sigma^2)$, the posterior distribution under

---

**Algorithm 1** The GP-MW algorithm for player $i$

---

**Input:** Set of actions $\mathcal{A}^i$, GP prior $(\mu_0, \sigma_0, k^i)$, parameters $\{\beta_t\}_{t \geq 1}, \eta$

1: Initialize: $\mathbf{w}_1^i = \frac{1}{K_i}(1, \ldots, 1) \in \mathbb{R}^{K_i}$
2: **for** $t = 1, 2, \ldots, T$ **do**
3:    Sample action $a_t^i \sim \mathbf{w}_t^i$
4:    Observe noisy reward $\tilde{r}_t^i$ and opponents' actions $a_t^{-i}$:

$$\tilde{r}_t^i = r^i(a_t^i, a_t^{-i}) + \epsilon_t^i$$

5:    Compute optimistic reward estimates $\hat{\mathbf{r}}_t \in \mathbb{R}^{K_i}$ :

$$[\hat{\mathbf{r}}_t]_a = \min\{1, UCB_t(a, a_t^{-i})\} \quad \text{for every} \quad a = 1, \ldots, K_i \quad (5)$$

6:    Update mixed strategy:

$$[\mathbf{w}_{t+1}^i]_a = \frac{[\mathbf{w}_t^i]_a \exp(-\eta\,(1 - [\hat{\mathbf{r}}_t]_a))}{\sum_{k=1}^{K_i}[\mathbf{w}_t^i]_k \exp(-\eta\,(1 - [\hat{\mathbf{r}}_t]_k))} \quad \text{for every} \quad a = 1, \ldots, K_i \quad (6)$$

7:    Update $\mu_t, \sigma_t$ according to (2)-(3) by appending $(\mathbf{a}_t, \tilde{r}_t^i)$ to the history of play.
8: **end for**

---

a $\mathcal{GP}(0, k(\mathbf{a}, \mathbf{a}'))$ prior is also Gaussian, with mean and variance functions:

$$\mu_t(\mathbf{a}) = \mathbf{k}_t(\mathbf{a})^\top (\mathbf{K}_t + \sigma^2 \mathbf{I}_t)^{-1} \mathbf{y}_t \quad (2)$$

$$\sigma_t^2(\mathbf{a}) = k(\mathbf{a}, \mathbf{a}) - \mathbf{k}_t(\mathbf{a})^\top (\mathbf{K}_t + \sigma^2 \mathbf{I}_t)^{-1} \mathbf{k}_t(\mathbf{a}), \quad (3)$$

where $\mathbf{k}_t(\mathbf{a}) = [k(\mathbf{a}_j, \mathbf{a})]_{j=1}^t$, $\mathbf{y}_t = [y_1, \ldots, y_t]^\top$, and $\mathbf{K}_t = [k(\mathbf{a}_j, \mathbf{a}_{j'})]_{j,j'}$ is the kernel matrix.

At time $t$, an upper confidence bound on $f$ can be obtained as:

$$UCB_t(\mathbf{a}) := \mu_{t-1}(\mathbf{a}) + \beta_t \sigma_{t-1}(\mathbf{a}), \quad (4)$$

where $\beta_t$ is a parameter that controls the width of the confidence bound and ensures $UCB_t(\mathbf{a}) \geq f(\mathbf{a})$, for all $\mathbf{a} \in \mathcal{A}$ and $t \geq 1$, with high probability [23]. We make this statement precise in Theorem 1.

Due to the above regularity assumptions and feedback model, player $i$ can use the history of play $\{(\mathbf{a}_1, \tilde{r}_1^i), \ldots, (\mathbf{a}_{t-1}, \tilde{r}_{t-1}^i)\}$ to compute an upper confidence bound $UCB_t(\cdot)$ of the unknown reward function $r^i$ by using (4). In the next section, we present our algorithm that makes use of $UCB_t(\cdot)$ to simulate full information feedback.

## 3   The GP-MW Algorithm

We now introduce GP-MW, a novel no-regret bandit algorithm, which can be used by a generic player $i$ (see Algorithm 1). GP-MW maintains a probability distribution (or mixed strategy) $\mathbf{w}_t^i$ over $\mathcal{A}^i$ and updates it at every time step using a multiplicative-weight (MW) subroutine (see (6)) that requires full information feedback. Since such feedback is not available, GP-MW builds (in (5)) an *optimistic* estimate of the true reward of every action via the upper confidence bound $UCB_t$ of $r^i$. Moreover, since rewards are bounded in $[0, 1]$, the algorithm makes use of $\min\{1, UCB_t(\cdot)\}$. At every time step $t$, GP-MW plays an action $a_t^i$ sampled from $\mathbf{w}_t^i$, and uses the noisy reward observation $\tilde{r}_t^i$ and actions $a_t^{-i}$ played by other players to compute the updated upper confidence bound $UCB_{t+1}(\cdot)$.

In Theorem 1, we present a high-probability regret bound for GP-MW while all the proofs of this section can be found in the supplementary material. The obtained bound depends on the *maximum information gain*, a kernel-dependent quantity defined as:

$$\gamma_t := \max_{\mathbf{a}_1, \ldots, \mathbf{a}_t} \frac{1}{2} \log \det(\mathbf{I}_t + \sigma^{-2} \mathbf{K}_t).$$

It quantifies the maximal reduction in uncertainty about $r^i$ after observing outcomes $\{\mathbf{a}_j\}_{j=1}^t$ and the corresponding noisy rewards. The result of [23] shows that this quantity is sublinear in $T$, e.g., $\gamma_T = \mathcal{O}((\log T)^{d+1})$ in the case of $k_{SE}$, and $\gamma_T = \mathcal{O}\big(T^{\frac{d^2+d}{2\nu+d^2+d}} \log T\big)$ in the case of $k_{Matérn}$, where $d$ is the total dimension of the outcomes $\mathbf{a} \in \mathcal{A}$, i.e., $d = \sum_{i=1}^N d_i$.

**Theorem 1.** *Fix $\delta \in (0,1)$ and assume $\epsilon_t^i$'s are $\sigma_i$-sub-Gaussian with independence over time. For any $r^i$ such that $\|r^i\|_{k^i} \leq B$, if player $i$ plays actions from $\mathcal{A}_i$, $|\mathcal{A}_i| = K_i$, according to GP-MW with $\beta_t = B + \sqrt{2(\gamma_{t-1} + \log(2/\delta))}$ and $\eta = \sqrt{(8 \log K_i)/T}$, then with probability at least $1 - \delta$,*

$$R^i(T) = \mathcal{O}\left(\sqrt{T \log K_i} + \sqrt{T \log(2/\delta)} + B\sqrt{T\gamma_T} + \sqrt{T\gamma_T(\gamma_T + \log(2/\delta))}\right).$$

The proof of this theorem follows by the decomposition of the regret of GP-MW into the sum of two terms. The first term corresponds to the regret that player $i$ incurs with respect to the sequence of computed upper confidence bounds. The second term is due to not knowing the true reward function $r^i$. The proof of Theorem 1 then proceeds by bounding the first term using standard results from adversarial online learning [7], while the second term is upper bounded by using regret bounding techniques from GP optimization [23, 4].

Theorem 1 can be made more explicit by substituting bounds on $\gamma_T$. For instance, in the case of the squared exponential kernel, the regret bound becomes $R^i(T) = \mathcal{O}\left(\left((\log K_i)^{1/2} + (\log T)^{d+1}\right)\sqrt{T}\right)$. In comparison to the standard multi-armed bandit regret bound $\mathcal{O}(\sqrt{TK_i \log K_i})$ (e.g., [3]), this regret bound does not depend on $\sqrt{K_i}$, similarly to the ideal full information setting.

### The case of continuous action sets

In this section, we consider the case when $\mathcal{A}^i$ is a (continuous) compact subset of $\mathbb{R}^{d_i}$. In this case, further assumptions are required on $r^i$ and $\mathcal{A}_i$ to achieve sublinear regret. Hence, we assume a bounded set $\mathcal{A}_i \subset \mathbb{R}^{d_i}$ and $r^i$ to be Lipschitz continuous in $a^i$. Under the same assumptions, existing regret bounds are $\mathcal{O}(\sqrt{d_i T \log T})$ and $\mathcal{O}(T^{\frac{d_i+1}{d_i+2}} \log T)$ in the full information [18] and bandit setting [22], respectively. By using a discretization argument, we obtain a high probability regret bound for GP-MW.

**Corollary 1.** *Let $\delta \in (0,1)$ and $\epsilon_t^i$ be $\sigma_i$-sub-Gaussian with independence over time. Assume $\|r^i\|_k \leq B$, $\mathcal{A}_i \subset [0,b]^{d_i}$, and $r^i$ is L-Lipschitz in its first argument, and consider the discretization $[\mathcal{A}^i]_T$ with $|[\mathcal{A}^i]_T| = (Lb\sqrt{d_i T})^{d_i}$ such that $\|a - [a]_T\|_1 \leq \sqrt{d_i/T}/L$ for every $a \in \mathcal{A}^i$, where $[a]_T$ is the closest point to $a$ in $[\mathcal{A}^i]_T$. If player $i$ plays actions from $[\mathcal{A}^i]_T$ according to GP-MW with $\beta_t = B + \sqrt{2(\gamma_{t-1} + \log(2/\delta))}$ and $\eta = \sqrt{8d_i \log(Lb\sqrt{d_i T})/T}$, then with probability at least $1 - \delta$,*

$$R^i(T) = \mathcal{O}\left(\sqrt{d_i T \log(Lb\sqrt{d_i T})} + \sqrt{T \log(2/\delta)} + B\sqrt{T\gamma_T} + \sqrt{T\gamma_T(\gamma_T + \log(2/\delta))}\right).$$

By substituting bounds on $\gamma_T$, our bound becomes $R^i(T) = \mathcal{O}(T^{1/2}\text{polylog}(T))$ in the case of the SE kernel (for fixed $d$). Such a bound has a strictly better dependence on $T$ than the existing bandit bound $\mathcal{O}(T^{\frac{d_i+1}{d_i+2}} \log T)$ from [22]. Similarly to [22, 18], the algorithm resulting from Corollary 1 is not efficient in high dimensional settings, as its computational complexity is exponential in $d_i$.

## 4 Experiments

In this section, we consider random matrix games and a traffic routing model and compare GP-MW with the existing algorithms for playing repeated games. Then, we show an application of GP-MW to robust BO and compare it with existing baselines on a movie recommendation problem.

### 4.1 Repeated random matrix games

We consider a repeated matrix game between two players with actions $\mathcal{A}_1 = \mathcal{A}_2 = \{0, 1, \ldots, K-1\}$ and payoff matrices $A^i \in \mathbb{R}^{K \times K}, i = 1, 2$. At every time step, each player $i$ receives a payoff $r^i(a_t^1, a_t^2) = [A^i]_{a_t^1, a_t^2}$, where $[A^i]_{i,j}$ indicates the $(i,j)$-th entry of matrix $A^i$. We select $K = 30$ and generate 10 random matrices with $r^1 = r^2 \sim GP(0, k(\cdot, \cdot))$, where $k = k_{SE}$ with $l = 6$. We set the noise to $\epsilon_t^i \sim \mathcal{N}(0, 1)$, and use $T = 200$. For every game, we distinguish between two settings:

**Against random opponent.** In this setting, player-2 plays actions uniformly at random from $\mathcal{A}^2$ at every round $t$, while player-1 plays according to a no-regret algorithm. In Figure 1a, we compare the

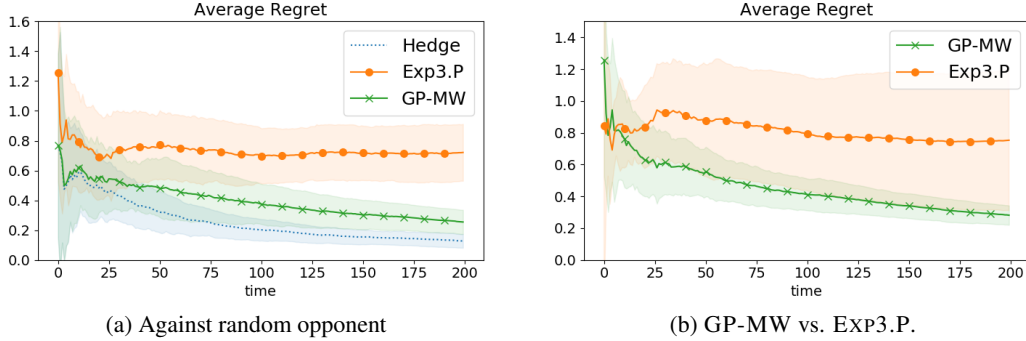

(a) Against random opponent       (b) GP-MW vs. EXP3.P.

Figure 1: GP-MW leads to smaller regret compared to EXP3.P. HEDGE is an idealized benchmark which upper bounds the achievable performance. Shaded areas represent ± one standard deviation.

time-averaged regret of player-1 when playing according to HEDGE [11], EXP3.P [3], and GP-MW. Our algorithm is run with the true function prior while HEDGE receives (unrealistic) noiseless full information feedback (at every round $t$) and leads to the lowest regret. When only the noisy bandit feedback is available, GP-MW significantly outperforms EXP3.P.

**GP-MW vs EXP3.P.** Here, player-1 plays according to GP-MW while player-2 is an adaptive adversary and plays using EXP3.P. In Figure 1b, we compare the regret of the two players averaged over the game instances. GP-MW outperforms EXP3.P and ensures player-1 a smaller regret.

### 4.2 Repeated traffic routing

We consider the Sioux-Falls road network [14, 1], a standard benchmark model in the transportation literature. The network is a directed graph with 24 nodes and 76 edges ($e \in E$). In this experiment, we have $N = 528$ agents and every agent $i$ seeks to send some number of units $u^i$ from a given origin to a given destination node. To do so, agent $i$ can choose among $K_i = 5$ possible routes consisting of network edges $E(i) \subset E$. A route chosen by agent $i$ corresponds to action $a^i \in \mathbb{R}^{|E(i)|}$ with $[a^i]_e = u^i$ in case $e$ belongs to the route and $[a^i]_e = 0$ otherwise. The goal of each agent $i$ is to minimize the travel time weighted by the number of units $u^i$. The travel time of an agent is unknown and depends on the total occupancy of the traversed edges within the chosen route. Hence, the travel time increases when more agents use the same edges. The number of units $u^i$ for every agent, as well as travel time functions for each edge, are taken from [14, 1]. A more detailed description of our experimental setup is provided in Appendix C.

We consider a repeated game, where agents choose routes using either of the following algorithms:

- HEDGE. To run HEDGE, each agent has to observe the travel time incurred had she chosen any different route. This requires knowing the exact travel time functions. Although these assumptions are unrealistic, we use HEDGE as an idealized benchmark.

- EXP3.P. In the case of EXP3.P, agents only need to observe their incurred travel time. This corresponds to the standard bandit feedback.

- GP-MW. Let $\psi(a_t^{-i}) \in \mathbb{R}^{|E(i)|}$ be the total occupancy (by other agents) of edges $E(i)$ at time $t$. To run GP-MW, agent $i$ needs to observe a noisy measurement of the travel time as well as the corresponding $\psi(a_t^{-i})$.

- Q-BRI (Q-learning Better Replies with Inertia algorithm [8]). This algorithm requires the same feedback as GP-MW and is proven to asymptotically converge to a Nash equilibrium (as the considered game is a potential game [19]). We use the same set of algorithm parameters as in [8].

For every agent $i$ to run GP-MW, we use a composite kernel $k^i$ such that for every $\mathbf{a}_1, \mathbf{a}_2 \in \mathcal{A}$, $k^i((a_1^i, a_1^{-i}), (a_2^i, a_2^{-i})) = k_1^i(a_1^i, a_2^i) \cdot k_2^i(a_1^i + \psi(a_1^{-i}), a_2^i + \psi(a_2^{-i}))$, where $k_1^i$ is a linear kernel and $k_2^i$ is a polynomial kernel of degree $n \in \{2, 4, 6\}$.

First, we consider a random subset of 100 agents that we refer to as learning agents. These agents choose actions (routes) according to the aforementioned no-regret algorithms for $T = 100$ game

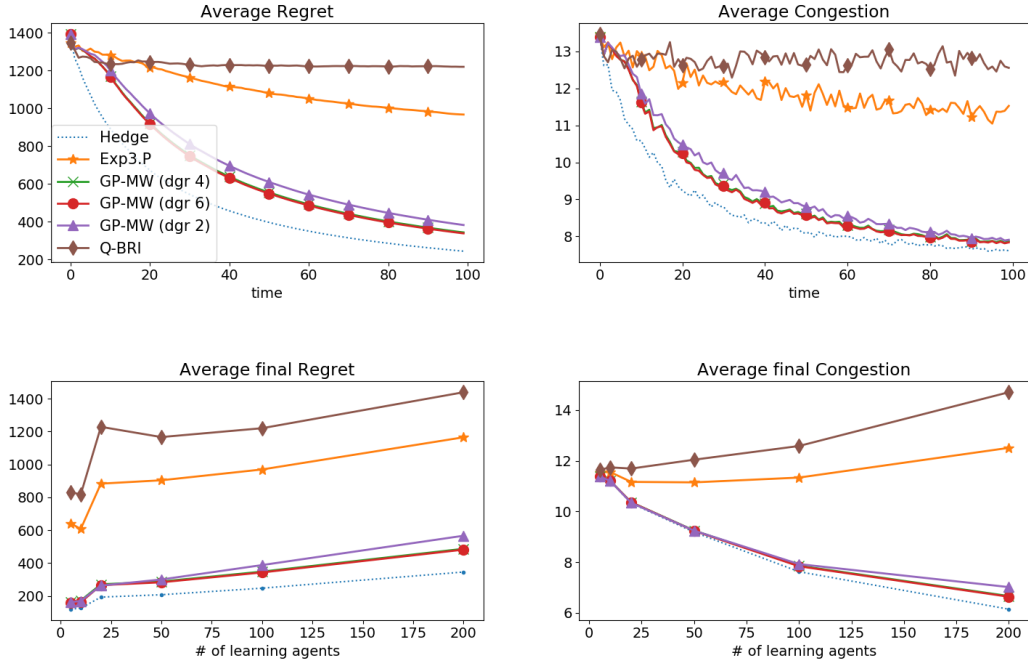

Figure 2: GP-MW leads to a significantly smaller average regret compared to EXP3.P and Q-BRI and improves the overall congestion in the network. HEDGE represents an idealized full information benchmark which upper bounds the achievable performance.

rounds. The remaining non-learning agents simply choose the shortest route, ignoring the presence of the other agents. In Figure 2 (top plots), we compare the average regret (expressed in hours) of the learning agents when they use the different no-regret algorithms. We also show the associated average congestion in the network (see (13) in Appendix C for a formal definition). When playing according to GP-MW, agents incur significantly smaller regret and the overall congestion is reduced in comparison to EXP3.P and Q-BRI.

In our second experiment, we consider the same setup as before, but we vary the number of learning agents. In Figure 2 (bottom plots), we show the final (when $T = 100$) average regret and congestion as a function of the number of learning agents. We observe that GP-MW systematically leads to a smaller regret and reduced congestion in comparison to EXP3.P and Q-BRI. Moreover, as the number of learning agents increases, both HEDGE and GP-MW reduce the congestion in the network, while this is not the case with EXP3.P or Q-BRI (due to a slower convergence).

### 4.3 GP-MW and robust Bayesian Optimization

In this section, we apply GP-MW to a novel robust Bayesian Optimization (BO) setting, similar to the one considered in [4]. The goal is to optimize an unknown function $f$ (under the same regularity assumptions as in Section 2) from a sequence of queries and corresponding noisy observations. Very often, the actual queried points may differ from the selected ones due to various input perturbations, or the function may depend on external parameters that cannot be controlled (see [4] for examples).

This scenario can be modelled via a two player repeated game, where a player is competing against an adversary. The unknown reward function is given by $f : \mathcal{X} \times \Delta \to \mathbb{R}$. At every round $t$ of the game, the player selects a point $x_t \in \mathcal{X}$, and the adversary chooses $\delta_t \in \Delta$. The player then observes the parameter $\delta_t$ and a noisy estimate of the reward: $f(x_t, \delta_t) + \epsilon_t$. After $T$ time steps, the player incurs the regret

$$R(T) = \max_{x \in \mathcal{X}} \sum_{t=1}^{T} f(x, \delta_t) - \sum_{t=1}^{T} f(x_t, \delta_t).$$

Note that both the regret definition and feedback model are the same as in Section 2.

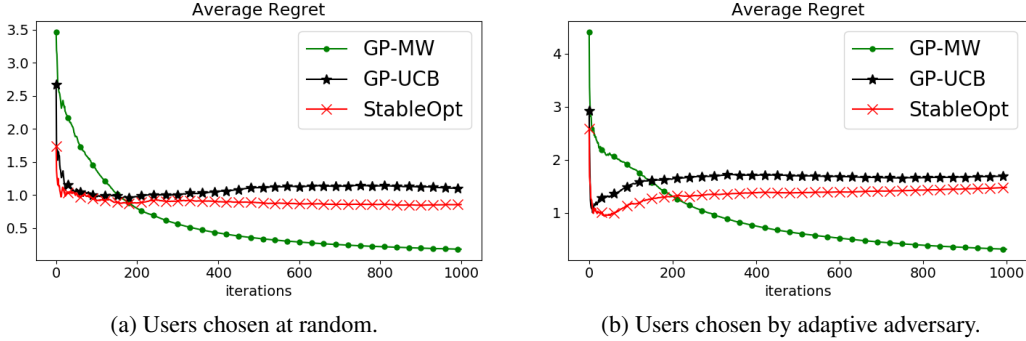

(a) Users chosen at random.

(b) Users chosen by adaptive adversary.

Figure 3: GP-MW ensures no-regret against both randomly and adaptively chosen users, while GP-UCB and STABLEOPT attain constant average regret.

In the standard (non-adversarial) Bayesian optimization setting, the GP-UCB algorithm [23] ensures no-regret. On the other hand, the STABLEOPT algorithm [4] attains strong regret guarantees against the worst-case adversary which perturbs the final reported point $x_T$. Here instead, we consider the case where the adversary is *adaptive* at every time $t$, i.e., it can adapt to past selected points $x_1, \ldots, x_{t-1}$. We note that both GP-UCB and STABLEOPT fail to achieve no-regret in this setting, as both algorithms are deterministic conditioned on the history of play. On the other hand, GP-MW is a no-regret algorithm in this setting according to Theorem 1 (and Corollary 1).

Next, we demonstrate these observations experimentally in a movie recommendation problem.

**Movie recommendation.** We seek to recommend movies to users according to their preferences. A priori it is unknown which user will see the recommendation at any time $t$. We assume that such a user is chosen arbitrarily (possibly adversarially), simultaneously to our recommendation.

We use the MovieLens-100K dataset [12] which provides a matrix of ratings for 1682 movies rated by 943 users. We apply non-negative matrix factorization with $p = 15$ latent factors on the incomplete rating matrix and obtain feature vectors $\mathbf{m}_i, \mathbf{u}_j \in \mathbb{R}^p$ for movies and users, respectively. Hence, $\mathbf{m}_i^\top \mathbf{u}_j$ represents the rating of movie $i$ by user $j$. At every round $t$, the player selects $\mathbf{m}_t \in \{\mathbf{m}_1, \ldots, \mathbf{m}_{1682}\}$, the adversary chooses (without observing $\mathbf{m}_t$) a user index $i_t \in \{1, \ldots, 943\}$, and the player receives reward $f(\mathbf{m}_t, i_t) = \mathbf{m}_t^\top \mathbf{u}_{i_t}$. We model $f$ via a GP with composite kernel $k((\mathbf{m}, i), (\mathbf{m}', i')) = k_1(\mathbf{m}, \mathbf{m}') \cdot k_2(i, i')$ where $k_1$ is a linear kernel and $k_2$ is a diagonal kernel.

We compare the performance of GP-MW against the ones of GP-UCB and STABLEOPT when sequentially recommending movies. In this experiment, we let GP-UCB select $\mathbf{m}_t = \arg\max_{\mathbf{m}} \max_i UCB_t(\mathbf{m}, i)$, while STABLEOPT chooses $\mathbf{m}_t = \arg\max_{\mathbf{m}} \min_i UCB_t(\mathbf{m}, i)$ at every round $t$. Both algorithms update their posteriors with measurements at $(\mathbf{m}_t, \hat{i}_t)$ with $\hat{i}_t = \arg\max_i UCB_t(\mathbf{m}_t, i)$ in the case of GP-UCB and $\hat{i}_t = \arg\min_i LCB_t(\mathbf{m}_t, i)$ for STABLEOPT. Here, $LCB_t$ represents a lower confidence bound on $f$ (see [4] for details).

In Figure 3a, we show the average regret of the algorithms when the adversary chooses users uniformly at random at every $t$. In our second experiment (Figure 3b), we show their performance when the adversary is adaptive and selects $i_t$ according to the HEDGE algorithm. We observe that in both experiments GP-MW is no-regret, while the average regrets of both GP-UCB and STABLEOPT do not vanish.

## 5  Conclusions

We have proposed GP-MW, a no-regret bandit algorithm for playing unknown repeated games. In addition to the standard bandit feedback, the algorithm requires observing the actions of other players after every round of the game. By exploiting the correlation among different game outcomes, it computes upper confidence bounds on the rewards and uses them to simulate unavailable full information feedback. Our algorithm attains high probability regret bounds that can substantially improve upon the existing bandit regret bounds. In our experiments, we have demonstrated the effectiveness of GP-MW on synthetic games, and real-world problems of traffic routing and movie recommendation.

**Acknowledgments**

This work was gratefully supported by Swiss National Science Foundation, under the grant SNSF 200021_172781, and by the European Union's Horizon 2020 ERC grant 815943.

## Footnotes

[1] In fact, they are allowed to be adaptive and adversarial.

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
