[Supplementary Material · supplementary_material.pdf]

# Supplementary Material

## No-Regret Learning in Unknown Games with Correlated Payoffs

**Pier Giuseppe Sessa, Ilija Bogunovic, Maryam Kamgarpour, Andreas Krause (NeurIPS 2019)**

## A   Proof of Theorem 1

We make use of the following well-known confidence lemma.

**Lemma 1** (Confidence Lemma). *Let $\mathcal{H}_k$ be a RKHS with underlying kernel function $k$. Consider an unknown function $f : \mathcal{A} \to \mathbb{R}$ in $\mathcal{H}_k$ such that $\|f\|_k \leq B$, and the sampling model $y_t = f(\mathbf{a}_t) + \epsilon_t$ where $\epsilon_t$ is $\sigma$-sub-Gaussian (with independence between times). By setting*

$$\beta_t = B + \sqrt{2(\gamma_{t-1} + \log(1/\delta))}$$

*the following holds with probability at least $1 - \delta$:*

$$|\mu_{t-1}(\mathbf{a}) - f(\mathbf{a})| \leq \beta_t \sigma_{t-1}(\mathbf{a}), \quad \forall \mathbf{a} \in \mathcal{A}, \quad \forall t \geq 1,$$

*where $\mu_{t-1}(\cdot)$ and $\sigma_{t-1}(\cdot)$ are given in (2)-(3).*

Lemma 1 follows directly from [2, Theorem 3.11 and Remark 3.13] as well as the definition of the maximum information gain $\gamma_{t-1}$.

We can now prove Theorem 1. Recall the definition of regret

$$R^i(T) = \max_{a \in \mathcal{A}^i} \sum_{t=1}^{T} r^i(a, a_t^{-i}) - \sum_{t=1}^{T} r^i(a_t^i, a_t^{-i}).$$

Defining $\bar{a} = \arg\max_{a \in \mathcal{A}^i} \sum_{t=1}^{T} r^i(a, a_t^{-i})$, $R^i(T)$ can be rewritten as

$$R^i(T) = \sum_{t=1}^{T} r^i(\bar{a}, a_t^{-i}) - \sum_{t=1}^{T} r^i(a_t^i, a_t^{-i}).$$

By Lemma 1 and since rewards are in $[0, 1]$, with probability $1 - \frac{\delta}{2}$ the true unknown reward function can be upper and lower bounded as:

$$UCB_t(\mathbf{a}) - 2\beta_t \sigma_{t-1}(\mathbf{a}) \leq r^i(\mathbf{a}) \leq \min\{1, UCB_t(\mathbf{a})\}, \quad \forall \mathbf{a} \in \mathcal{A}^1 \times \cdots \times \mathcal{A}^N, \quad \forall t \geq 1, \quad (7)$$

with $UCB_t$ defined in (4) and $\beta_t$ chosen according to Theorem 1. Thus, $UCB_t(\mathbf{a}) - 2\beta_t \sigma_{t-1}(\mathbf{a})$ is a lower confidence bound of $r^i(\mathbf{a})$.

Hence,

$$R^i(T) \leq \sum_{t=1}^{T} \min\{1, UCB_t(\bar{a}, a_t^{-i})\} - \sum_{t=1}^{T} \left[ UCB_t(a_t^i, a_t^{-i}) - 2\beta_t \sigma_{t-1}(a_t^i, a_t^{-i}) \right]$$

$$\leq \sum_{t=1}^{T} \min\{1, UCB_t(\bar{a}, a_t^{-i})\} - \sum_{t=1}^{T} \min\{1, UCB_t(a_t^i, a_t^{-i})\} + 2\beta_T \sum_{t=1}^{T} \sigma_{t-1}(a_t^i, a_t^{-i}),$$

where the first inequality follows by (7) and the second one since $\beta_t$ is increasing in $t$.

Moreover, by [23, Lemma 5.4] and the choice $\beta_T = B + \sqrt{2(\gamma_T + \log(2/\delta))}$, we have

$$2\beta_T \sum_{t=1}^{T} \sigma_{t-1}(a_t^i, a_t^{-i}) = \mathcal{O}\left( B\sqrt{T\gamma_T} + \sqrt{T\gamma_T(\gamma_T + \log(2/\delta))} \right).$$

Next, we show that with probability $1 - \frac{\delta}{2}$,

$$\sum_{t=1}^{T} \min\{1, UCB_t(\bar{a}, a_t^{-i})\} - \sum_{t=1}^{T} \min\{1, UCB_t(a_t^i, a_t^{-i})\}$$

$$= \mathcal{O}\left( \sqrt{T \log K_i} + \sqrt{T \log(2/\delta)} \right). \quad (8)$$

The statement of the theorem then follows by standard probability arguments:

$$\mathbb{P}[E_1 \cap E_1] = \mathbb{P}[E_1] + \mathbb{P}[E_2] - \mathbb{P}[E_1 \cup E_2] \geq \left(1 - \frac{\delta}{2}\right) + \left(1 - \frac{\delta}{2}\right) - 1 = 1 - \delta \,,$$

where $E_1$ and $E_2$ are the events (7) and (8), respectively.

To show (8), define the function $f_t^i(\cdot) = \min\{1, UCB_t(\cdot, a_t^{-i})\}$. Note that if (7) holds, $UCB_t(\cdot) \geq 0$ since $r^i(\cdot) \geq 0$, hence $f_t^i(\cdot) \in [0,1]^{K_i}$. Using such definition, the left hand side of (8) can be upper bounded as:

$$\sum_{t=1}^T f_t^i(\bar{a}) - \sum_{t=1}^T f_t^i(a_t^i) \leq \max_{a \in \mathcal{A}^i} \sum_{t=1}^T f_t^i(a) - \sum_{t=1}^T f_t^i(a_t^i) \,. \tag{9}$$

Observe that the right hand side of (9) is precisely the regret which player $i$ incurrs in an adversarial online learning problem with reward functions $f_t^i(\cdot) \in [0,1]$. The actions $a_t^i$, moreover, are exactly chosen by the HEDGE [11] algorithm which receives the full information feedback $\hat{\mathbf{r}}_t = [f_t^i(a_1), \ldots, f_t^i(a_{K_i})]$. Note that the original version of HEDGE works with losses instead of rewards, but the same happens in GP-MW since the mixed strategies are updated with $\mathbf{1} - \hat{\mathbf{r}}_t$. Therefore, by [7, Corollary 4.2], with probability $1 - \frac{\delta}{2}$,

$$\max_{a \in \mathcal{A}^i} \sum_{t=1}^T f_t^i(a) - \sum_{t=1}^T f_t^i(a_t^i) = \mathcal{O}\left(\sqrt{T \log K_i} + \sqrt{T \log(2/\delta)}\right) \,.$$

Note that according to [7, Remark 4.3], the functions $f_t^i(\cdot)$ can be chosen by an adaptive adversary depending on past actions $a_1^i, \ldots, a_{t-1}^i$, but not on the current action $a_t^i$. This applies to our setting, since $f_t^i$ depends only on $a_1^i, \ldots, a_{t-1}^i$ and not on $a_t^i$. $\qquad\square$

## B  Proof of Corollary 1

A function $f : \mathcal{X} \to \mathbb{R}$ is Lipschitz continuous with constant $L$ (or $L$-Lipschitz) if

$$|f(x) - f(x')| \leq L\|x - x'\|_1 \qquad \forall x, x' \in \mathcal{X} \,.$$

Define $\mathcal{A}^{-i} = \mathcal{A}^1 \times \cdots \times \mathcal{A}^{i-1} \times \mathcal{A}^i \times \cdots \times \mathcal{A}^N$. The fact that $r^i$ is $L$-Lispschitz in its first argument implies that

$$|r^i(a, a^{-i}) - r^i(a', a^{-i})| \leq L\|a - a'\|_1 \qquad \forall a, a' \in \mathcal{A}^i, \forall a^{-i} \in \mathcal{A}^{-i} \,. \tag{10}$$

Moreover, recall the discrete set $[\mathcal{A}^i]_T$ with $|[\mathcal{A}^i]_T| = (Lb\sqrt{d_i T})^{d_i}$ such that $\|a - [a]_T\|_1 \leq bd_i/Lb\sqrt{d_i T} = \sqrt{d_i/T}/L \,\forall a \in \mathcal{A}^i$, where $[a]_T$ is the closest point to $a$ in $[\mathcal{A}^i]_T$. An example of such a set can be obtained for instance by a uniform grid of points in $[0, b]^{d_i}$.

As in the proof of Theorem 1, let $\bar{a} = \arg\max_{a \in \mathcal{A}^i} \sum_{t=1}^T r^i(a, a_t^{-i})$. Moreover, let $[\bar{a}]_T$ be the closest point to $\bar{a}$ in $[\mathcal{A}^i]_T$. We have:

$$R^i(T) = \sum_{t=1}^T r^i(\bar{a}, a_t^{-i}) - \sum_{t=1}^T r^i(a_t^i, a_t^{-i})$$

$$= \underbrace{\sum_{t=1}^T r^i(\bar{a}, a_t^{-i}) - \sum_{t=1}^T r^i([\bar{a}]_T, a_t^{-i})}_{:=R_1^i(T)} + \underbrace{\sum_{t=1}^T r^i([\bar{a}]_T, a_t^{-i}) - \sum_{t=1}^T r^i(a_t^i, a_t^{-i})}_{:=R_2^i(T)} \,.$$

We prove the corollary by bounding $R_1^i(T)$ and $R_2^i(T)$ separately.

By the Lipschitz property (10) of $r^i$, and by construction of $[\mathcal{A}^i]_T$, we have that

$$|r^i(\bar{a}, a^{-i}) - r^i([\bar{a}]_T, a^{-i})| \leq L\|\bar{a} - [\bar{a}]_T\|_1 \leq L\frac{\sqrt{d_i/T}}{L} = \sqrt{d_i/T}, \qquad \forall a^{-i} \in \mathcal{A}^{-i} \,. \tag{11}$$

Hence, by (11),

$$R_1^i(T) \leq T\sqrt{d_i/T} = \sqrt{d_i T} \,.$$

To bound $R_2^i(T)$, note that $R_2^i(T) \leq \arg\max_{a \in [\mathcal{A}^i]_T} \sum_{t=1}^{T} r^i(a, a_t^{-i}) - \sum_{t=1}^{T} r^i(a_t^i, a_t^{-i})$. Moreover, note that actions $a_t^i$ are chosen by running GP-MW on the discretized domain $[\mathcal{A}^i]_T$ with $K_i = |[\mathcal{A}^i]_T| = (Lb\sqrt{d_iT})^{d_i}$. Hence, according to Theorem 1 it must hold that with probability at least $1 - \delta$,

$$R_2^i(T) = \mathcal{O}\left( \sqrt{T \log K_i} + \sqrt{T \log(2/\delta)} + B\sqrt{T\gamma_T} + \sqrt{T\gamma_T(\gamma_T + \log(2/\delta))} \right).$$

The final bound then follows by substituting $K_i = (Lb\sqrt{d_iT})^{d_i}$ in the bound above and noting that $R_1^i(T)$ is dominated by $R_2^i(T)$. $\qquad\square$

## C   Repeated traffic routing - Experimental setup

In this section we give a detailed explanation of our traffic routing experiment of Section 4.2.

We consider the Sioux-Falls road network [14, 1], a directed graph with 24 nodes and 76 edges $e \in E$. We use the demand data from [14, 1]. Such data indicate the units of flow to be sent from each node (origin) to any other node (destination) in the network. Each of those origin-destination pair is here represented by an agent, for a total of $N = 528$ agents. The goal of each agent $i$ is to send $u^i$ units of demand to destination, while minimizing the total travel time. The time to reach destination, however, depends on the total occupancy of the edges the agent chooses to traverse and hence on the routes chosen by all the other agents.

Each edge $e$ has a travel time $t_e(x)$ which is a function of the total number of units $x$ traversing $e$. Intuitively, we expect such travel time to increase with $x$. According to [14, 1], we select $t_e$ to be the Bureau of Public Roads (BPR) function

$$t_e(x) = c_e\left(1 + 0.15\left(\frac{x}{C_e}\right)^4\right),$$

where $c_e$ and $C_e$ are free-flow time and capacity of edge $e$, respectively. Values for $c_e$ and $C_e$ are taken from [1].

Each agent $i$ can choose among $K^i = 5$ routes, and we assume that she cannot split her demand over different routes. Hence, the action space $\mathcal{A}^i$ represents the 5 shortest routes that agent $i$ can take. Moreover, we remove from $\mathcal{A}^i$ any route more than three times longer than the shortest one. Let $E(i) \subset E$ be the subset of edges that agent $i$ could possibly traverse. Each route in $\mathcal{A}^i$ corresponds to a vector $a^i \in \mathbb{R}^{|E(i)|} \in \mathcal{A}^i$ such that $[a^i]_e = u^i$ if edge $e$ belongs to the given route, and $[a^i]_e = 0$ otherwise. Moreover, we let $\psi(a^{-i}) = \in \mathbb{R}^{|E(i)|}$ be the total occupancy by the other agents on such edges, i.e., $[\psi(a^{-i})]_e = \sum_{j \neq i}[a^j]_e$ for every $e \in E(i)$. The travel time of agent $i$ can thus be written as

$$l^i(a^i, a^{-i}) = \sum_{e \in E(i)} [a^i]_e \, t_e([a^i]_e + [\psi(a^{-i})]_e), \tag{12}$$

i.e., the sum of the travel times on the selected edges, weighted by $u^i$. Hence, we let the reward function of agent $i$ be $r^i(a^i, a^{-i}) = -l^i(a^i, a^{-i})$.

Note that agents don't know the actual $t_e$'s functions, hence their reward function is unknown. This does not limit the bandit EXP3.P algorithm, where agents only need to observe their experienced travel times. However, it makes the full information feedback HEDGE algorithm unrealistic. Nevertheless, we used HEDGE in our experiments as an idealized benchmark.

To run GP-MW, agent $i$ observes the experienced travel time as well as the vector of occupancies $\psi(a^{-i})$. This allows GP-MW to exploit the correlations in the unknown reward function by choosing a suitable kernel. For every agent $i$, we chose a composite kernel $k^i$ such that for every $\mathbf{a}_1, \mathbf{a}_2 \in \mathcal{A}$, $k^i((a_1^i, a_1^{-i}), (a_2^i, a_2^{-i})) = k_1^i(a_1^i, a_2^i) \cdot k_2^i(a_1^i + \psi(a_1^{-i}), a_2^i + \psi(a_2^{-i}))$, with $k_1^i$ and $k_2^i$ being linear and polynomial kernels, respectively. This reflects the different dependences that $r^i$ has on $a^i$ and $a^{-i}$. In fact, for fixed total occupancy in each edge, we expect $r^i$ to be linear in $a^i$, being the travel time an additive quantity (see (12)). On the other hand, given a specific route chosen, $r^i$ grows polynomially with the total occupancy on such route (see (12)). Kernels hyperparameters are optimized via maximum-likelihood over 200 random outcomes.

To scale their rewards in [0,1] agents need to know upper bounds on their travel times. Such bounds are estimated by $10'000$ random outcomes and fed to the agents. Moreover, standard deviations of

measurement noises are chosen 0.1 % of such upper bounds. Finally, to evaluate a given outcome $\mathbf{a}_t$ of the game, we compute the congestion on a given edge $e$ via the expression:

$$0.15 \cdot \Big( \sum_{j=1}^{N} [a_t^j]_e / C_e \Big)^4 . \tag{13}$$

The average congestion in the network is obtained by averaging the quantity above over all the edges $e \in E$.