[Reviews · NeurIPS 2019]

Reviewer 1



The paper considers learning in the following model: the player repeatedly plays in the same game, that is not known a-priory. Gets feedback only about the his value for the realized outcome, but also learns actions of the other players. Two important assumptions are made about the game: 1. a regularity assumption that the game has a low dimensional kernel, and 2. that the response function is noisy with Gaussian noise on top of the originally unknown value. The paper offers an elegant algorithm combining Kernelized multiarm bandit algorithms (from ICML10 and ICML'17) and multiplicative weights to get a no-regret strategy. The main idea is the use the UCB-style upper bounds in a multiplicative weight algorithm, which moves the performance of the algorithm closer to full information feedback. Section 4 of the paper offers empirical evidence on the better performance of the algorithm on three different classes of examples. Unfortunately a number of the important details of these examples are deferred to supplementary materials. But there is still a convincing case for the improved performance of the algorithm. I read the authors response. The response provided for my question (of what application can the model proposed be a good model for) is not great. True, that maybe one can get information on the number of agents traveling a route, but the more natural sources of information is actually the delay there (say offered by Google maps). And true, one can observe properties of others bidding in the auction space, maybe true for the winning bidder, but not all participants, and not all their action. I view the paper as a first step in an interesting direction, hence my relatively high rating of 7.

Reviewer 2



EDIT after rebuttal: The paper proposes a new feedback model which is new. However, the algorithmic idea and analysis is not new. This is probably the weakest point of the paper. So it boils down to if the question is itself useful (e.g., has useful applications). I agree that 7 is a generous score. My primary reason was that this seemed like an interesting direction: to understand how much can the full-feedback be relaxed while getting the same bounds. The stated motivations in the rebuttal are pretty weak; in fact, even if feasible I would not release the number of agents traveling a route due to privacy issues (e.g., if a route is sensitive and telling if > 0 agents choosing a route gives away too much information). On the other hand, I thought that robust bayesian optimization, in the paper, seemed like a reasonable one. --- This paper considers the no-regret learning dynamics in multi-player game. They consider the setting where one player is using a no-regret algorithm that receives a noisy reward for the chosen action and the actions chosen by the other players (but not their rewards). They further assume a regularity assumption on the rewards that is required by their solution which uses gaussian process (GP) to estimate UCB on the rewards of the other players. The key result is to illustrate that under this restrictive feedback, in many applications they are able to get as fast a convergence rate as receiving full feedback and always faster than bandit feedback. They argue that this is a natural feedback that is received in many applications and full-feedback is restrictive for applications. The propose an algorithm, prove regret bounds and evaluate this on simulated and real-world transportation dataset. My comments for this paper as follows. I think the feedback model they consider and the goal of the paper is novel. It is interesting to understand how much feedback one really needs to get fast bounds and the trade-off between feedback and convergence rate is a research worth pursuing. The algorithm seems to be a rather simple modification of the usual exponential weights approach after appropriately estimating the rewards of the other players from their actions using standard tools from GP. Thus, the algorithm and hence the analysis seems to be a straight forward marriage between the techniques in these two areas. But, they provide *extensive* experimental evaluation on simulated and real-world transportation datasets. They also show an application in Bayesian optimization previously considered in prior work and compare their algorithms with that in prior work. The lack of significant novelty in theory is made up by the extensive experimental evaluation. As a bonus, this paper is very well-written and easy to read. Weighing the pros and cons, I would lean towards acceptance of this paper. I had a question to the authors. In the bandit setting and 2-player games, when both players play no-regret algorithms faster convergence rates can be obtained (for instance [1]). Is something similar possible for this model of feedback? In summary, the pros and cons are as follows: Pros: - Novel feedback model - Extensive experimental evaluation - Paper extremely well-written Cons: - Algorithm and analysis is rather straightforward once the problem is correctly setup. [1] - Fast convergence of regularized learning in games - Syrgkanis, Agarwal, Luo, Schapire, NIPS 2015.

Reviewer 3



The paper is well-written and technically sound. As mentioned below, the significance is less clear as the kernel assumption is not well justified. Detailed comments: (1) Page 2, Table 1: Just based on what is in the table, it is really unclear how GP-MW improves over EXP3. The setting of EXP3 and GP-MW are different in several aspects and it is unclear which one of the differences causes the difference in regret bound. For example, what is the optimal regret bound of EXP3's setting plus kernel assumption? And what is the optimal regret bound of GP-MW's setting without the kernel's assumption? To make a fair comparison between EXP3 and GP-MW, the authors can also talk about how GP-MW performs when all opponents have only 1 action to choose from and there is no kernel assumption. (2) The authors do not spend enough effort to justify why the kernel assumption makes sense. Do most forms of games have this property? Not everyone is an expert on this assumption and it is really hard to judge the result without further justification of the assumption. (3) In the regret definition in this paper, a_t^{-i} is a fixed sequence. This mean the paper does not consider that other players are using adaptive strategies and chosen actions can affect other players' actions. It would be better to make this clear. (4) Line 170: Is the improvement mainly because of the kernel assumption? (5) In the experiments, although some part of the data is from real-world, the rewards are generated using kernels. So the experiments also don't justify the kernel assumption. I have read the author response. "The Kernel assumption is standard in kernelized literature" does not sound like a good justification for the assumption. I agree that it is technically convenient. I would hope the authors can provide more justification from game theory perspective, i.e. showing examples of games which are standard in game theory and work well with the kernel assumption.

[Author Response · NeurIPS 2019]

We would like to thank all the reviewers for their constructive feedback. In the following, we respond (**R**) to individual concerns (**C**) summarized in italic. Citations refer to references in the paper and to the additional ones provided below.

**Reviewer 1. C:** *"I do agree that full information feedback is hard to expect in real scenarios,... However, the current model ...is just as unreasonable to expect. Is there an application where this is a more realistic assumption?"*
**R:** The main motivation for our model is a setting that is in between the full information and bandit feedback. In such settings, opponents' actions, or some similar aggregate information, can often be observed in practice. In the considered routing application (Section 4.2), the number of agents that traverse routes can be monitored and made available to the agents, while the delay functions are in general unknown to the agents. Also, when recommending items (Section 4.3), users' preferences are unknown, but one can typically observe the profile of the user who rates the recommendation. The proposed feedback model is also present in other practical applications. For example, in repeated auctions, bidders do not know their reward functions since these functions depend on complex auction-clearing mechanisms, but can often observe their payoffs as well as some aggregate information about the opponents' bids [26]. In competitions between firms in a shared market, each firm sets a price and subsequently observes its experienced revenues as well as the prices set by the other firms [27]. Finally, the feedback required by our model is natural in *cooperative* multi-agent systems where the agents communicate their actions to each other.

**C:** *"What is the regularity parameter $\gamma_T$ in the routing example?"* **R:** Computing the *kernel-dependent* quantity $\gamma_T$ is in general a hard problem, and bounds only exist for most widely-used kernels [23]. To the best of our knowledge, no bounds exist for the used polynomial kernels. We note that we observed comparable experimental performance when we switched to the Squared Exponential (SE) kernel (for which a bound on $\gamma_T$ exists).

**Reviewer 2. C:** *"...when both players play no-regret algorithms faster convergence rates can be obtained. Is something similar possible for this model of feedback?"* **R:** Regardless of other players' strategies, the individual regret of GP-MW always contains an additive $\mathcal{O}(\sqrt{T})$ term due to learning the unknown reward function. Therefore, differently from [24], an overall convergence rate of $\mathcal{O}(\sqrt{T})$ seems inevitable.

**C:** *"It is interesting to understand how much feedback one really needs to get fast bounds and the trade-off between feedback and convergence rate is a research worth pursuing."* **R:** We agree that the trade-off between feedback and convergence rate is a research worth pursuing and we see our work as a contribution in this direction.

**Reviewer 3. C:** *"The authors do not spend enough effort to justify why the kernel assumption makes sense. Do most forms of games have this property?"* **R:** Assuming the reward function $r^i(\cdot)$ has a bounded RKHS norm with respect to some kernel $k$, is a standard assumption used in kernelized stochastic bandit literature (e.g., [23], [9], etc.) to effectively enforce smoothness on the reward function. This leads to the fact that similar game outcomes would produce similar rewards, and allows a player to use the observed history of play to learn about $r^i(\cdot)$ and generalize for unseen game outcomes. Note that our results are not restricted to any specific kernel function, and depending on the application at hand, various kernels can be used to model different types of reward functions. We will elaborate more on the made assumption both formally and intuitively in our paper.

We answer to the remaining questions of Reviewer 3 using the same numbering provided in the review:
**(1):** The intention of Table 1 was to summarize the regret bounds of algorithms that require *different feedback*. We will clarify this aspect more in the paper. Exp3 algorithm works in the standard multi-armed bandit setting and it does not exploit potentially present correlations between outcomes and rewards. Moreover, we are not aware of sharper regret bounds for Exp3 when rewards are *noisy* and satisfy the kernel assumption. Furthermore, it is not obvious how to modify Exp3 to make use of the additional feedback considered in GP-MW. On the other hand, under the proposed feedback model, GP-MW is able to exploit such correlations and achieve a kernel-dependent regret bound. Additionally, in the case in which rewards are not correlated (corresponding to a diagonal kernel), the constant $\gamma_T$ in Theorem 1 grows as $\mathcal{O}(K_i)$ and hence, our approach does not provide any improvement over the regret bound of Exp3.
**(3):** The obtained regret guarantees also hold under *adaptive* opponents' strategies. We mentioned this in Footnote 1 and proof of Theorem 1. We will further clarify this in the paper.
**(4):** Yes, the improvement in Line 170 arises from the kernel assumption which allows emulating the full information feedback guarantees, similarly as in Theorem 1.
**(5):** This is true when it comes to our first experiment, while this is **not** the case in our real-world experiments (Sections 4.2, 4.3). For example, in 4.2, the reward function is obtained via the Bureau of Public Roads congestion model [14]. GP-MW is then run with a polynomial kernel whose parameters are learned from observed data. We experimented with different kernels and found out that similar results can also be obtained with the Squared Exponential (SE) kernel (which has universal function approximation properties and is typically a default choice when no additional domain knowledge is available).

**Additional References**

[26] J. Weed, V. Perchet, and P. Rigollet. Online learning in repeated auctions. *COLT*, 2016.
[27] U. Nadav and G. Piliouras. No Regret Learning in Oligopolies: Cournot vs. Bertrand. *Algorithmic Game Theory*, 2010.


[Meta-Review · NeurIPS 2019]

Reviewers are lukewarm, after a discussion. Pros: achieving full-feedback-like regret under only partial feedback and some regularity assumptions is intellectually important. This result also forms an interesting counterpoint with [SALS]. Cons: insufficient motivation for the feedback model and for the kernel assumption. Also, the analysis appears fairly standard. [SALS] Vasilis Syrgkanis, Alekh Agarwal, Haipeng Luo, and Robert E. Schapire. Fast convergence of regularized learning in games. NIPS 2015. --------------- Some additional points: The kernel assumption is a very strong assumption which has been used in a line of prior work on "Gaussian Process bandits" as a way to formalize the idea that "similar arms bring similar rewards". AFAIK, this work does not claim that this assumption has much to do with reality. Instead, they claim it is productive, in the sense that their algorithms work well in some semi-idealized experiments, where the data is *not* generated according to their model. It would serve the authors to explain the novelty of their technical approach. The main technical idea seems to be, to plug in UCBs into an EXP3-like algorithm, and analyze this algorithm by decomposing regret into the terms coming from the UCB estimators and the terms coming from multiplicative weights. However, these ideas have been used in several papers, and trace back to EXP3.P algorithm (the high-confidence version of EXP3 in the original paper). The application to "robust Bayesian optimization" could use some more motivation. To fit it into the main model, the impact of the "environment" is modeled as observable action delta_t such that the reward function is smooth in delta_t (in the sense of the kernel assumption). But it is really unclear whether/when anything like this happens. Also, only one paper is cited, so (smoothness assumption aside) it is unclear if this setting is standard / well-studied.